# Effect of Wearable Exoskeleton Robots on Muscle Activation and Gait Parameters on a Treadmill: A Randomized Controlled Trial

**DOI:** 10.3390/healthcare13070700

**Published:** 2025-03-22

**Authors:** Kyung-Jin Lee, Yeon-Gyo Nam, Jae-Ho Yu, Jin-Seop Kim

**Affiliations:** Department of Physical Therapy, Sun Moon University, Asan-si 31460, Republic of Korea; leekyungjin0510@gmail.com (K.-J.L.); naresa@sunmoon.ac.kr (J.-H.Y.); skylove3373@sunmmon.ac.kr (J.-S.K.)

**Keywords:** maximal voluntary contraction, muscle fatigue, stance phase, swing phase, loading response, pre-swing, step length, stride length

## Abstract

**Background:** Exoskeleton robots are emerging as a transformative technology in healthcare, rehabilitation, and industrial settings, providing significant benefits such as improving gait restoration and preventing injuries. These robots enhance mobility for individuals with neuromuscular disorders by providing muscular assistance and reducing physical strain, while also supporting workers in physically demanding tasks. They improve gait efficiency, muscle activation, and overall physical function, contributing to both rehabilitation and occupational health. **Objective:** This study aims to investigate the impact of exoskeleton use on muscle activation patterns, fatigue levels, and gait parameters in healthy individuals. **Methods:** Thirty-six participants engaged in a randomized sequence gait experiment on a treadmill for 30 min, both with and without an exoskeleton, with electromyography (EMG) and OptoGait measurements collected during the sessions. A one-week washout period was implemented before participants switched conditions. **Results:** In the Maximum voluntary contraction (MVC) analysis, significant differences were observed in the Rectus femoris (RF) and gastrocnemius(GM) when wearing the exoskeleton robot compared to not wearing it. At 10 min, 20 min, and 30 min, the differences were statistically significant (*p* < 0.05) for all muscles. In the muscle fatigue analysis, significant differences were observed in RF, GM, vastus medialis (VM), and hamstring(HS) at 10 min, 20 min, and 30 min (*p* < 0.05). In the step length and stride length analysis, significant differences were observed at 10 min and 30 min, but no differences were found at 20 min (*p* < 0.05). **Conclusions:** This study demonstrates that the use of the exoskeleton robot significantly impacts muscle activation, muscle fatigue, and gait parameters. The results emphasize the potential benefits of exoskeletons in enhancing mobility and reducing muscle strain, providing important insights for rehabilitation and occupational applications

## 1. Introduction

Exoskeleton robots are emerging as a transformative technology in healthcare and rehabilitation, with significant potential to improve and restore gait ability [1]. This technology is becoming increasingly relevant due to various social needs, including the protection of workers’ health in industrial settings, the growing demand for rehabilitation after accidents or illnesses, and the treatment of neuromuscular and musculoskeletal disorders.

In industrial environments, exoskeletons are used to reduce physical strain and enhance productivity, helping workers avoid musculoskeletal injuries caused by repetitive or heavy tasks [2,3]. Individuals with mobility limitations due to accidents or medical conditions benefit from exoskeletons, which provide valuable muscular assistance, facilitate rehabilitation, and enable more independent movement in daily life. This advancement, driven by the convergence of healthcare and robotics, offers a new path toward improving the mobility and quality of life of those with physical disabilities [4].

Traditional methods often rely on manual therapy or therapist-guided exercises, which may lack the intensity or consistency required for optimal recovery. Exoskeleton robots provide precise movements that can enhance muscle activation, improve gait efficiency, and reduce physical strain during movement [5,6,7]. They assist individuals with limited mobility by activating specific muscle groups, maintaining balance, and helping them practice walking, all of which contribute significantly to restoring gait patterns and muscle memory [8,9].

Patients with spinal cord injuries, stroke, or muscle atrophy can experience improved walking stability with the support of an exoskeleton, which helps the body adjust to natural walking motions [10,11]. This technology enhances opportunities for users to engage in their physical functions, gradually supporting the recovery of physical independence.

Occupational exoskeletons are wearable devices designed to assist workers in physically demanding tasks within various job settings [12]. These exoskeletons have the potential to lessen physical strain and alleviate the challenges of manual labor, which may help minimize productivity losses and mitigate adverse socioeconomic effects. In recent years, numerous exoskeleton designs, both active and passive, have been developed and assessed for their effectiveness [13].

Long-term use of an exoskeleton in real-world work environments requires a thorough investigation into the accumulation of muscle fatigue, variations in muscle activation, and the adaptation of gait patterns. Understanding the physical changes that occur due to prolonged exoskeleton use is significant, especially in relation to how the device’s operational mechanisms affect the user’s body over time. Such analysis helps monitor muscle fatigue levels, identify excessive activation of specific muscle groups, and detect abnormal changes in gait patterns. By examining these physical responses, we can better define the key design elements that should be considered in the development process.

This research can contribute to improving the design and application strategies of exoskeletons, particularly in industrial settings, leading to more efficient and safer usage. By understanding the long-term effects of exoskeleton use on muscles and proposing technical solutions to reduce any negative impacts, this study can enable workers and rehabilitation patients to use exoskeletons more comfortably and effectively. This will help minimize physical strain on workers and enhance gait improvement in rehabilitation, playing a key role in reducing injury risk and improving functional outcomes.

## 2. State of the Art

### 2.1. Maximum Voluntary Contraction with Wearable Exoskeleton

Previous studies have demonstrated that wearable exoskeletons effectively reduce muscle activation and redistribute loads across muscle groups, particularly in relation to MVC. Kong et al. (2022) reported that exoskeleton use decreases lower limb muscle activity, including the biceps femoris, rectus femoris (RF), and tibialis anterior, as measured by EMG [14]. Similarly, Shim et al. (2023) found that RF, biceps femoris, and tibialis anterior activity significantly decreased when wearing an exoskeleton at different working heights [15].

Pillai (2020) highlighted that leg support exoskeletons reduce RF muscle activity during both static and dynamic squatting tasks, further emphasizing their role in decreasing muscle demand [16]. These findings suggest that exoskeletons contribute to reduced muscle load, potentially preventing injuries by alleviating muscle fatigue and promoting more efficient movement.

Most previous studies have focused on short-term effects, such as single gait cycles or a few minutes of movement [14,15]. However, this study extends the analysis to 10, 20, and 30-min walking sessions, providing valuable insights into muscle fatigue and adaptation over time. Examining how different muscles compensate for one another when wearing an exoskeleton allows for a deeper understanding of lower limb coordination patterns.

### 2.2. Muscle Fatigue Analysis with Wearable Exoskeleton

Wearable exoskeletons have been extensively studied for their impact on muscle activation and fatigue during treadmill walking [17,18]. EMG data has been widely used to assess changes in muscle engagement, offering precise insights into how exoskeletons assist in reducing fatigue.

Wang (2021) reported that an exoskeleton in follow mode does not restrict natural movement and effectively reduces muscle fatigue [19]. Van Sluijs et al. (2023) developed a method to quantify reductions in back and hip muscle fatigue when using lift-support exoskeletons, reinforcing the importance of exoskeleton assistance in physically demanding tasks [20]. Chang (2023) found that wearing an ultralight hip exoskeleton resulted in an 83.5% decrease in tibialis anterior fatigue, along with significant reductions in RF, semitendinosus, and gastrocnemius activity. Additionally, oxygen uptake and energy expenditure per minute were reduced by 14% and 12.9%, respectively, suggesting metabolic benefits [21].

Bosch (2015) reported that exoskeleton reduced lower back muscle activity by 35–38% during assembly tasks and increased endurance in static holding tasks, extending task duration from 3.2 min to 9.7 min while maintaining comfort levels [22].

Through EMG analysis, researchers can identify variations in muscle engagement patterns, revealing the extent to which exoskeletons influence muscle fatigue, support, and efficiency during prolonged walking sessions. This highlights the importance of optimizing exoskeleton assistance to enhance muscle function while minimizing physical strain.

### 2.3. Changes in Gait Parameters Associated with Exoskeleton

Several studies have investigated the impact of wearable exoskeletons on gait parameters, including step length, stride length, and phase duration [21,23]. These analyses provide insight into users’ adaptive responses to exoskeleton support and how the body adjusts to assisted gait.

Farkhatdinov (2018) found that an exoskeleton robot improved gait patterns by increasing hip and knee movement, reducing step duration, and enhancing RF and biceps femoris activation without restricting natural gait [24]. Similarly, Lee et al. (2017) reported that robotic-assisted gait increased gait speed, stride length, and single support time while reducing RF and medial gastrocnemius activity in the terminal stance and pre-swing phases [25].

This study expands upon previous research by simultaneously analyzing muscle activation, fatigue, and gait parameters over an extended period. While existing studies have primarily examined muscle activation [26,27,28], few have comprehensively evaluated both gait parameters and muscle fatigue, including MVC. By addressing this gap, this research provides a more holistic understanding of how exoskeletons influence walking mechanics and muscle function over time.

### 2.4. Research Summary and Implications

Wearable exoskeletons have been extensively studied for their effects on muscle activation, fatigue reduction, and gait adaptation. Existing research demonstrates that exoskeletons reduce muscle activity, alleviate fatigue, and influence gait parameters, suggesting their potential benefits in occupational and rehabilitation settings. Studies on MVC show that exoskeletons help redistribute muscular loads and minimize strain on lower limb muscles [14,15,16]. Additionally, research on muscle fatigue confirms that exoskeletons can delay fatigue onset, optimize energy expenditure, and enhance endurance [19,20]. Investigations into gait parameters highlight how exoskeletons influence stride patterns, step length, and movement efficiency without restricting natural gait.

Despite these advancements, most prior studies have focused on short-term effects, limiting the understanding of muscle adaptation and fatigue over extended durations [20,21,22]. Additionally, research has primarily concentrated on isolated parameters rather than examining the combined effects of muscle activation, fatigue, and gait simultaneously. Notably, no study has comprehensively analyzed all three aspects together within a single experimental framework.

This study addresses these gaps by simultaneously assessing the effects of exoskeleton use on MVC, muscle fatigue, and gait parameters over an extended 30-min walking session. By integrating these three key components into a single study, this research provides a more holistic understanding of how exoskeletons influence human movement over time. By examining these factors together, this study contributes to the optimization of exoskeleton technology for both rehabilitation and occupational applications. The findings highlight the importance of a multidimensional approach in exoskeleton research, as considering all three aspects together provides a clearer perspective on their overall effects and benefits.

## 3. Evaluating Exoskeletons for Industrial Use: Purpose and Impact

This research aims to assess how these advancements improve gait patterns and reduce muscle fatigue, offering a comprehensive understanding of the potential of exoskeletons in supporting human mobility. This study evaluates the applicability of exoskeletons in industrial settings, assessing their ability to reduce muscle fatigue and improve walking efficiency during prolonged use. By focusing on functional support rather than rehabilitation, this research provides valuable insights into how exoskeletons can enhance workplace safety and performance, potentially preventing work-related musculoskeletal disorders.

## 4. Materials and Methods

### 4.1. Participants

A total of 36 young adults volunteered to participate in this study (Table 1). They had no history of lower extremity or lower-back surgery and were free from physical injuries in the past 6 months. Individuals with cardiovascular, neurological disease, or conditions that could hinder the maximal application of effort were excluded. The study was conducted with 36 participants, each undergoing the process twice, with a one-week interval between sessions.

Written informed consent was obtained from all participants after explaining the study purpose and experimental design. They were also aware of their right to withdraw from this study. The Institutional Review Board of DONGBANG CULTURE University approved this study (8223-202410-HR-013-01), which was performed in accordance with the principles of the Declaration of Helsinki [29].

### 4.2. Experimental Procedure

A total of 36 participants completed a treadmill walking session at a constant speed of 1.7 km/h in 30 min. The experiment consisted of two conditions: walking with and without the exoskeleton. These conditions were alternated within the same session, ensuring identical walking parameters for comparison.

To analyze muscle activation, fatigue, and gait parameters, EMG and OptoGait measurements were collected simultaneously throughout the walking session. The EMG electrodes were placed on four key muscles—RF along the midline of the thigh, GM on the upper calf, HS on the posterior thigh, and VM on the inner thigh just above the knee—following standard placement guidelines [30,31]. Gait parameters were recorded using the OptoGait system, which was positioned along the treadmill. The system captured stance phase, swing phase, loading response, pre-swing, step length, and stride length data over a 3 m segment of the treadmill belt, excluding the first and last 2 m to avoid acceleration and deceleration effects. Both EMG and gait parameters were measured every 10 min during the 30 min session to assess changes in muscle activation, fatigue, and walking mechanics over time.

To ensure reliability, the same participants repeated the experiment one week later, walking under the same conditions while alternating between the exoskeleton and non-exoskeleton trials. The exoskeleton was designed to provide external support, and all experimental parameters—including treadmill speed and measurement intervals—were kept consistent across trials to allow for direct comparison (Figure 1 and Figure 2).

### 4.3. Exoskeleton Robot

The LEXO-V Wearable Robot (Hancom, Yuseong-gu, Daejeon, Republic of Korea) is a non-powered strength-assist exoskeleton designed to reduce muscle strain and support walking. This device provides external support to enhance mobility and prevent injuries in both industrial and rehabilitation settings [27]. The LEXO-V Wearable Robot used in this study is designed to assist with walking by providing external support to reduce muscle load and enhance mobility. It features motors and actuators that help facilitate movement, allowing the wearer to walk more efficiently while minimizing fatigue. It is worn during gait analysis to measure its effects on muscle activation in participants. This wearable robot is a non-powered strength-assisting exoskeleton that is comfortable and easy to wear, like clothing. It supports the wearer’s lower limb strength, making it easier and more convenient to lift heavy objects. The device firmly supports the waist, shoulders, and legs, promoting proper posture and helping prevent injuries. The exoskeleton was adjusted to fit each participant individually. It reduces working time by 33%, decreases metabolic energy consumption by 32%, and improves work endurance. Additionally, it helps prevent musculoskeletal injuries. The product weighs 2 kg, supports a maximum load of 30 kg, and is adjustable to different sizes. The front of the device is equipped with a shoulder harness, chest harness, and a wire lock/release switch, while the rear features a spine support, waist harness, wire fixation module, thigh harness, and gloves (Figure 3).

### 4.4. Electromyography

In this study, EMG data were collected using the FREEEMG 1000 system (BTSG-sensor, made in Songpa-gu, Seoul, Republic of Korea), with the data transferred to a personal computer using MyoResearch XP Master 1.06 software. The system was configured with a sampling rate of 1024 Hz, a band-pass filter between 80–250 Hz, and a notch filter at 60 Hz to reduce noise. The EMG system measured MVC and muscle fatigue. The MVC trial was normalized using each selected muscle group’s peak muscle activation level and expressed as a percentage of MVC [32]. Median frequency (MDF) values can be obtained through frequency spectrum analysis. Among these, MDF is the most used indicator of muscle fatigue [33]. The fatigue index was calculated by the following Formula (1). The intra-rater reliability of the EMG measurements in this study was 0.83, indicating good consistency across trials [34].

Maximum Voluntary Contraction (MVC) is expressed as a percentage of the maximum EMG signal recorded during a reference contraction. Specifically, each participant performed an MVC test before the experiment, and the peak EMG amplitude obtained during this test was used as a normalization factor. The EMG signals recorded during walking trials were then expressed as a percentage of this MVC value (%MVC), allowing for standardized comparisons across participants and conditions.

Muscle fatigue was assessed based on the root mean square (RMS) amplitude and the median frequency (MF) shift of the EMG signal over time. A progressive decrease in MF and an increase in RMS amplitude indicate muscle fatigue. The RMS values are reported in microvolts (µV), while the MF is expressed in Hertz (Hz). To quantify changes over time, we analyzed muscle fatigue at 10, 20, and 30 min during the walking trials.
(1)$$\mathrm{Fatigue}\;(\%)=\frac{{MDF}_{before}-{MDF}_{after}}{{MDF}_{before}}\;\times \;100\%$$

### 4.5. Optogait

Optogait system (Microgate made in Bolzano, Italy) was used as a gait measurement device. This system consists of two 4 m bars for transmission and reception as well as a Logitech Webcam Pro 9000 (Logitech made in San Jose, CA, USA). for visual monitoring. The bars contain 1 cm LEDs that communicate via infrared and foot sensors that capture data on gait parameters. The participants walked on a treadmill at a constant speed of 1.7 km/h for 30 min. OptoGait, consisting of a 7-m-long sensor bar, was positioned along the treadmill to capture gait parameters. Data were collected over a 3 m segment of the treadmill belt, excluding the first and last 2 m to avoid acceleration and deceleration effects.

The gait parameters measured using the OptoGait system include stance phase, swing phase, loading response, pre-swing, step length, and stride length. For parameters expressed as percentages (%), these values represent the proportion of the total gait cycle occupied by each respective phase. Stance phase, swing phase, loading response, and pre-swing are all expressed as percentages of the total gait cycle.

For parameters expressed in centimeters (cm), such as step length and stride length, we provide reference values for normal gait based on existing literature. In healthy adults, typical step length ranges from 60 to 80 cm, while stride length ranges from 120 to 160 cm. Including these reference values allows for a comparative interpretation of the gait characteristics observed in this study.

The system demonstrated high reliability, with intra- and inter-rater reliability (ICC) values ranging from 0.88 to 0.95 [35].

### 4.6. Statistical Analysis

All measured values are expressed as mean and standard deviation. A paired *t*-test was conducted to compare the differences in muscle activation and gait parameters between the conditions of walking with and without the exoskeleton robot. The level of statistical significance was set as *p* < 0.05. All procedures were performed using SPSS software (SPSS 27.0, Armonk, NY, USA).

## 5. Results

### 5.1. Comparison of Maximum Voluntary Contraction Values During Gait with and Without an Exoskeleton

RF, a significant difference was observed with a value of (*p* = 0.014) at 10 min, and a value of (*p* < 0.01) at both 20 and 30 min (*p* < 0.05). GM, a significant difference was observed with a value of (*p* < 0.01) at both 10 and 20 min, and (*p* = 0.001) at 30 min (*p* < 0.05). HS, a significant difference was observed with a value of (*p* = 0.001) at 20 min and a value of (*p* < 0.01) at 30 min (*p* < 0.05). VM, a significant difference was observed with a (*p* = 0.001) at 20 min and a value of (*p* < 0.01) at 30 min (*p* < 0.01) (Table 2).

As shown in Table 2, the exoskeleton condition resulted in a significantly lower MVC compared to the non-wear condition after 30 min of walking. This supports the idea that exoskeletons progressively decrease muscle activation, reducing fatigue and enabling more efficient movement. In contrast, walking without exoskeleton led to greater muscle fatigue, resulting in higher MVC values over time.

Table 2 presents the reduction in GM activation at 10 and 20 min during exoskeleton use, likely due to the external support provided by the device. However, by 30 min, fatigue accumulated in the exoskeleton condition, leading to decreased GM activation compared to the non-exoskeleton condition. This shift indicates that, over time, the exoskeleton’s passive support might become less effective as fatigue sets in, requiring the muscles to compensate more.

Table 2 shows that the changes in HS MVC can be attributed to muscle fatigue, increased reliance on the exoskeleton for support over time, and reduced muscle activation as the device takes over some of the load. Initially, there is little difference, but as time progresses, the exoskeleton’s support becomes more pronounced, leading to a significant reduction in muscle activation. The HS coordinates knee flexion and hip extension during walking.

As the exoskeleton supports the lower body, it reduces the load on the VM, leading to less activation for both movements. This allows the VM to work less, which helps explain the reduced muscle activation during longer durations of walking with the exoskeleton.

### 5.2. Comparison of Muscle Fatigue Values During Gait with and Without Exoskeleton

For the RF, a difference was observed with a *p*-value of 10 min (*p* < 0.01), 20 min (*p* < 0.05) and 30 min (*p* < 0.01). GM and HS, significant differences were observed at 10, 20, and 30 min with (*p* < 0.01). A significant difference in the VM was observed between 10 and 30 min (*p* < 0.01) and at 20 min (*p* < 0.05) (Table 3).

Table 3 shows that the progressive increase in muscle fatigue over time can be attributed to factors such as muscle fatigue accumulation and adaptation, particularly when the exoskeleton is not worn. As the activity continues, fatigue may accumulate, requiring the muscle to exert more effort to maintain performance, which leads to higher levels of fatigue.

### 5.3. Comparison of Stance Phase and Swing Phase Values During Gait with and Without Exoskeleton

During the stance phase, a difference was observed in the right foot, with (*p* = 0.01) at 10 min, (*p* = 0.012) at 20 min, and (*p* < 0.01) at 30 min. With the left foot, significant differences were observed at 10, 20, and 30 min (*p* < 0.01). In the swing phase, a difference was observed in the right foot, with (*p* = 0.001) at 10 min and (*p* < 0.01) at 30 min. For the left foot, significant differences were observed at 10, 20, and 30 min (*p* < 0.05) (Table 4).

Table 4 shows that wearing the Exoskeleton influences the muscle activation and force of the lower body muscles, particularly during the stance phase. In the early stages of the stance phase, the Exoskeleton provided significant support, likely aiding in better weight distribution and stability, which could explain the reduction in muscle activation. As time progressed, however, the support from the Exoskeleton might have diminished, potentially due to muscular fatigue, discomfort, or a decrease in the device’s effectiveness. This adaptation could have altered the user’s gait mechanics, resulting in reduced muscle activation and force (Table 4).

### 5.4. Comparison of Loading Response and Pre-Swing Values During Gait with and Without Exoskeleton

In terms of loading response, a difference was observed in the right foot, with (*p* < 0.01). In the left foot, significant differences were observed, with (*p* = 0.001) at 10 min and both 20 and 30 min (*p* < 0.01). Pre-swing phase, the right foot showed a difference, with (*p* = 0.049) at 10 min and (*p* < 0.01) at both 20 and 30 min. In the left foot, significant differences were observed, with (*p* = 0.041) at 20 min and (*p* < 0.01) at 30 min (*p* < 0.05) (Table 5).

In the pre-swing phase, there was no noticeable difference in the left foot between the 10-min and 20-min marks. However, for the right foot, there were differences at 10, 20, and 30 min, with the differences becoming more noticeable as time went on (Table 5). In the loading response phase, the values without the exoskeleton were higher at the 10 min mark, but gradually decreased as time progressed (Table 5).

### 5.5. Comparison of Step Length and Stride Length Measurements During Gait with and Without Exoskeleton Support

Both step length and stride length, significant differences were observed at 10, and 30 min (*p* < 0.05). The step length showed a significant difference (*p* < 0.01) in both the left and right feet after walking for 10 min. Similarly, after 30 min of walking, the step length also showed a significant difference (*p* < 0.01) in both feet. The stride length exhibited a significant difference (*p* < 0.01) after 10 min of walking, and a significant difference (*p* < 0.01) was also observed in the stride length after 30 min (Table 6).

Table 6 shows significant differences in step length and stride length between the 10 min and 30 min time points; however, no significant differences were noted at the 20 min mark. This phenomenon may be attributed to several factors related to gait adaptation, fatigue progression, and compensatory mechanisms. Participants may have required an initial adaptation period to adjust to the exoskeleton and treadmill walking conditions. During the first 10 min, adjustments in step length and stride length could have resulted from this adaptation phase. By the 20 min mark, participants may have reached a steady-state gait pattern, leading to an absence of significant differences. However, as fatigue accumulated by the 30 min mark, changes in gait parameters became more pronounced again.

## 6. Discussion

### 6.1. LEXO-V Wearable Robot

The LEXO-V Wearable Robot, developed as a non-powered strength-assist exoskeleton, is designed to support users in tasks requiring prolonged standing and walking. Unlike motorized exoskeletons, the LEXO-V relies on a passive assist mechanism that enhances stability and reduces muscle strain during movement.

The device features a lightweight and robust backplate to ensure stability while preventing restriction of the lower back during bending motions. It incorporates a Kevlar-based wire system with a tensile strength exceeding 150 kg, providing secure load distribution. A key component of its design is the wire length adjustment device, which utilizes a pulley stop mechanism synchronized with the end effector, allowing for adaptive support based on the user’s movement. The LEXO-V is worn around the waist and thighs, providing external support that minimizes fatigue accumulation and excessive muscle activation. The exoskeleton assists with maintaining optimal posture and reducing biomechanical stress, particularly in industrial and occupational settings. Through its ergonomic design, it facilitates ease of movement while enhancing work efficiency and injury prevention.

### 6.2. Muscle Activation, Fatigue Accumulation, and Gait Adaptation During Extended Exoskeleton Use

This study investigated the effects of exoskeleton use on muscle activation and gait parameters during walking. The results indicate that wearing an exoskeleton led to distinct changes in muscle activation patterns and influenced key gait parameters. These results suggest that exoskeleton support can positively impact neuromuscular engagement and promote gait stability, aligning with previous research on the potential benefits of exoskeletons in reducing muscle fatigue and enhancing mobility.

The results of the study by Su-Hyun Lee et al. (2023) emphasize the potential benefits of wearable assistive technologies, such as the exoskeleton robot, in improving mobility and physical function in elderly individuals. Similarly, our study demonstrates the effects of wearable assistive devices in enhancing gait, reducing muscle fatigue, and improving physical function in elderly participants [36].

Research has shown that wearing an exoskeleton significantly reduces muscle activation during gait, and the results of this study support these conclusions [16,37]. Previous studies have demonstrated that exoskeleton use decreases muscle activity in both the upper and lower limbs compared to walking without one [16]. Similarly, a decrease in RF activity has been observed, ranging from 12% to 57%, depending on the specific task performed [22]. These results align with previous research [22,37], which suggest that exoskeletons enhance endurance and reduce physiological strain on muscles.

These results align with those of Raasch and Zajac (1999), who observed similar effects during cycling. The ankle joint utilizes different strategies, such as dorsiflexion and plantarflexion, to generate and transfer power, storing and releasing kinetic energy [38]. This suggests that the external support provided by the exoskeleton may reduce the need for such active muscular interventions at the ankle joint, which could explain the reduced activation of the GM muscle in the early stages of the task.

Prolonged walking without exoskeleton (Exo) support led to greater fatigue in the GM, likely due to its crucial role in propulsion and load absorption during gait. As a biarticular muscle, GM not only contributes to ankle plantarflexion during push-off, but also plays a stabilizing role at the knee joint [39]. Without Exo assistance, the muscle must generate greater force to maintain propulsion, leading to increased metabolic demand and a higher rate of neuromuscular fatigue accumulation.

HS activity peaked at 95% of the gait cycle [40], and with exoskeleton use, HS fatigue progressively decreased over the 10, 20, and 30 min walking periods. The slight increase in HS activity in the powered conditions compared to unpowered ones suggests that the exoskeleton reduced the muscle’s workload, helping to delay fatigue and improve endurance during prolonged walking.

Baptista et al. (2022) suggested that walking overground and on a treadmill with a powered lower-limb exoskeleton induces muscle fatigue in the knee flexors [41]. To prevent excessive fatigue in the HS, additional exercises targeting these muscles should be avoided in the same training session. That result supports this study, which demonstrated that muscle fatigue was reduced when wearing the exoskeleton compared to not wearing it.

Previous research has shown that muscle fatigue leads to a decrease in MVC over time, indicating reduced force-generating capacity [30]. In our study, we observed that, when wearing the exoskeleton, MVC gradually decreased in the VM, suggesting that the device reduced muscular demand.

Conversely, without the exoskeleton, MVC tended to increase, which may indicate compensatory muscle activation to counteract fatigue. Muscle fatigue is typically associated with a reduction in MVC over prolonged exertion [30]. Given that MVC decreased in the VM when using the exoskeleton, it is likely that the device helped reduce muscle fatigue by offloading the muscle workload during prolonged walking.

This supports the idea that external support can alleviate muscle strain and delay fatigue accumulation. Studies have reported that prolonged exertion can lead to compensatory muscle activation to maintain force output despite ongoing fatigue [39]. In our experiment, the increase in MVC in VM without the exoskeleton suggests that participants may have engaged in a compensatory response to sustain performance over time.

Previous studies, such as [42], observed an increase in muscle activation and force overtime when the Exoskeleton was worn. The present study demonstrated a reduction in activation and force for the knee extensors in the early stance phase. This discrepancy may be attributed to differences in experimental protocols, such as the duration of the Exoskeleton wear and the level of muscle fatigue experienced by the user.

Muscle fatigue does not always increase in a linear fashion. Instead, there may be periods where compensatory mechanisms temporarily stabilize performance. Between 10 and 20 min, participants might have engaged muscle groups more efficiently to counteract early fatigue effects. However, as fatigue progressively accumulated, these compensatory mechanisms may have become less effective, leading to significant gait alterations at 30 min. Furthermore, individuals tend to alter their gait strategies in response to fatigue and external support. At a fatigue level of 20 min, participants may have adjusted their walking mechanics to maintain step length and stride length. These adjustments could include optimizing joint kinematics, redistributing muscular effort, or modifying cadence to sustain efficiency. By 30 min, these adjustments might no longer have been sufficient, resulting in measurable differences. The absence of significant differences in step length and stride length at 20 min is likely influenced by the combined effects of adaptation, non-linear fatigue patterns, and compensatory gait strategies. Understanding these dynamics aids in optimizing exoskeleton design and enhancing user performance during prolonged walking sessions.

Previous research by Hultman et al. has shown that during prolonged exercise, such as a 30 min walking session, glycogenolysis occurs at a low rate, resulting in sustained force production despite gradual glycogen depletion, which may influence performance and fatigue levels [43]. A duration of 30 min provides enough time for metabolic and physiological changes, such as shifts in motor unit recruitment and energy efficiency, to take place. It also reflects practical applications where workers or individuals using exoskeletons would engage in sustained walking tasks [44].

Gender-related differences in muscle activation, fatigue resistance, and gait biomechanics could influence the effects of the exoskeleton. Men generally have greater muscle mass and different neuromuscular control strategies, while women tend to exhibit greater fatigue resistance and joint flexibility, which may lead to differences in how the exoskeleton interacts with the body. These factors could influence load distribution, muscular effort, and overall gait adaptation when wearing the device. However, this study did not include a sex-based analysis because the primary focus was on the overall effects of wearing the LEXO-V exoskeleton rather than individual differences.

## 7. Conclusions

This study demonstrates that the use of the exoskeleton robot significantly impacts muscle activation, muscle fatigue, and gait parameters. The results emphasize the potential benefits of exoskeletons in enhancing mobility and reducing muscle strain, providing important insights for rehabilitation and occupational applications. However, a limitation of this study is its focus on healthy adults, which may not fully capture the potential benefits of exoskeletons for more diverse populations. Additionally, the relatively short duration of the intervention suggests that further research with longer intervention periods is needed to fully understand the exoskeleton’s potential for rehabilitation and performance enhancement.

Future research should investigate the effects of longer intervention periods to fully understand the exoskeletons’ potential for rehabilitation and performance enhancement. Moreover, since the current data is limited to healthy adults, expanding research to various age groups and populations could provide a more comprehensive insights.

## Figures and Tables

**Figure 1 healthcare-13-00700-f001:**
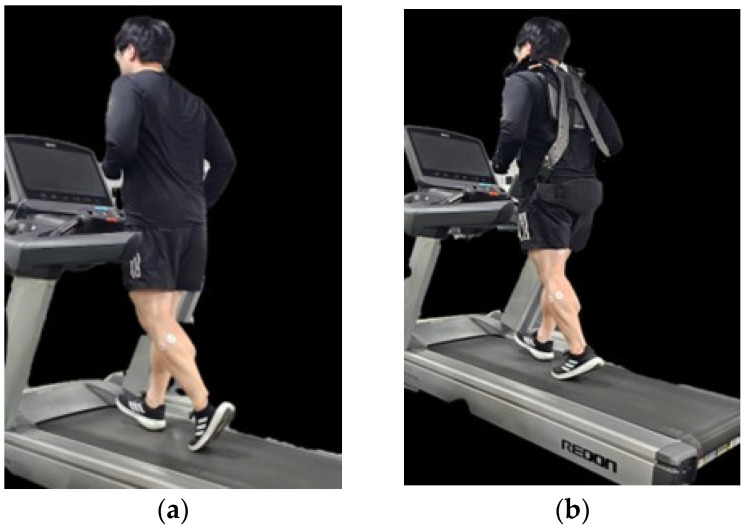
(**a**) Gait on a treadmill without the exoskeleton robot (**b**) Gait on a treadmill with the exoskeleton robot.

**Figure 2 healthcare-13-00700-f002:**
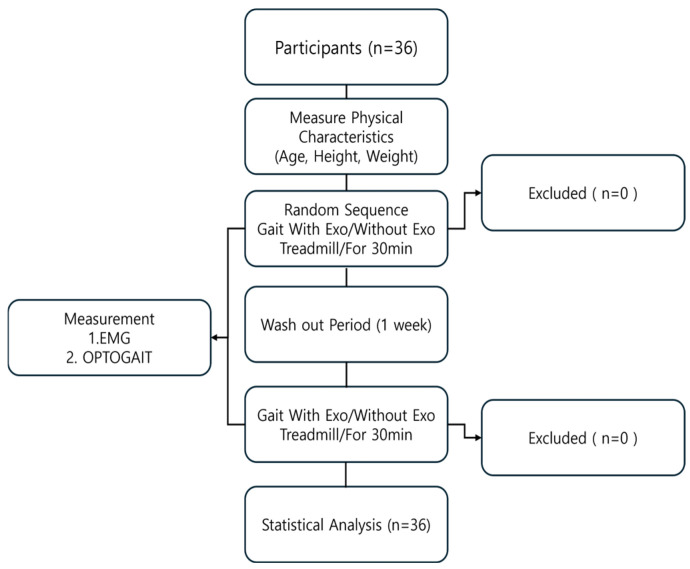
Flow diagram for Experiment.

**Figure 3 healthcare-13-00700-f003:**
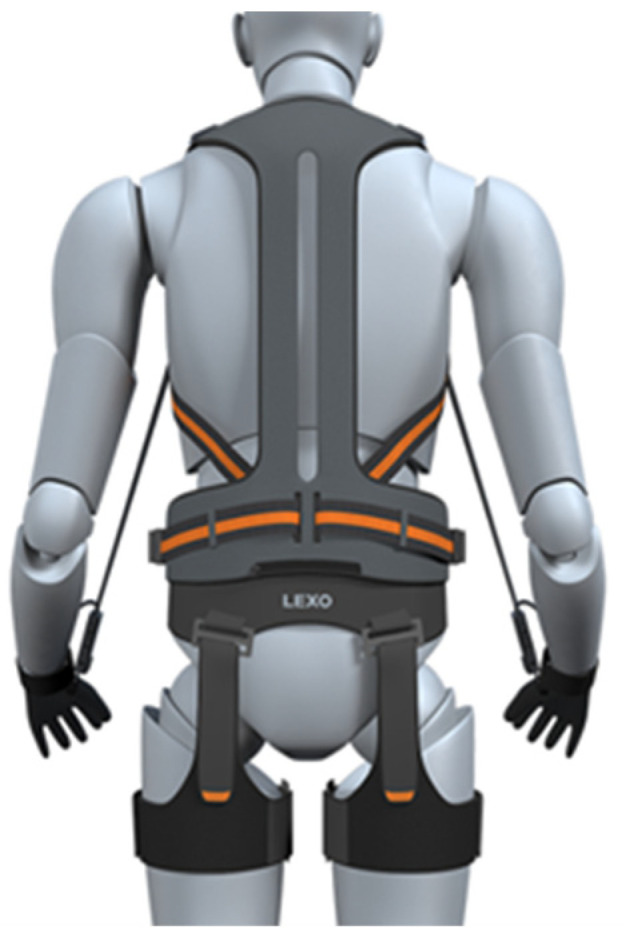
Exoskeleton robot (LEXO-V Wearable Robot).

**Table 1 healthcare-13-00700-t001:** Physical characteristics of the participants.

Variables	Total (n = 36)
Age	23.10 ± 4.70
Sex	Male, n = 21, Female, n = 15
Height, cm	168.92 ± 8.68
Body weight, kg	63.99 ± 10.43
Body mass index, kg/m^2^	22.48 ± 2.51

Data are presented as mean ± standard deviation.

**Table 2 healthcare-13-00700-t002:** Maximum voluntary contraction values during gait with and without exoskeleton.

MVC (%)		10 min	t	*p*	20 min	t	*p*	30 min	t	*p*
Rectus femoris	W/O Exo	44.93 ± 9.92	2.576	0.014 *	46.49 ± 8.66	5.241	<0.00 **	52.39 ± 5.70	10.946	<0.00 **
	W/Exo	40.60 ± 3.39			36.7 ± 3.32			32.81 ± 3.57		
Gastrocnemius medialis	W/O Exo	18.25 ± 3.18	−11.58	<0.00 **	19.46 ± 3.42	−6.936	<0.00 **	24.34 ± 4.39	3.711	0.001 *
	W/Exo	24.73 ± 2.16			23.59 ± 2.11			21.33 ± 2.25		
Hamstring	W/O Exo	17.05 ± 3.61	0.069	0.945	19.22 ± 3.96	3.595	0.001 *	24.16 ± 4.13	10.329	<0.00 **
	W/Exo	17.00 ± 2.05			16.49 ± 2.06			14.16 ± 3.41		
Vastus medialis	W/O Exo	26.70 ± 2.95	−0.657	0.515	19.22 ± 3.96	3.651	0.001 *	36.60 ± 6.13	13.36	<0.00 **
	W/Exo	27.21 ± 3.35			25.19 ± 2.14			22.21 ± 3.43		

** p* < 0.05; ** *p* < 0.01. Data are presented as mean ± standard deviation. W/O Exo: without exoskeletons, W/Exo: with exoskeletons.

**Table 3 healthcare-13-00700-t003:** Muscle fatigue values during gait with and without exoskeleton.

Muscle Fatigue (%)		10 min	t	*p*	20 min	t	*p*	30 min	t	*p*
Rectus Femoris	W/O Exo	29.88 ± 8.74	−6.552	<0.00 **	34.22 ± 7.36	2.326	0.026 *	42.90 ± 7.07	8.284	<0.00 **
	W/Exo	39.26 ± 2.97			31.32 ± 4.13			32.08 ± 4.97		
gastrocnemius medialis	W/O Exo	23.56 ± 4.71	2.480	<0.00 **	28.18 ± 3.83	14.097	<0.00 **	33.60 ± 4.53	18.702	<0.00 **
	W/Exo	21.32 ± 3.38			18.93 ± 1.84			16.13 ± 3.32		
Hamstring	W/O Exo	25.14 ± 3.92	−9.088	<0.00 **	30.57 ± 2.49	3.491	<0.00 **	37.22 ± 3.17	20.096	<0.00 **
	W/Exo	32.25 ± 3.40			27.64 ± 5.06			24.91 ± 2.01		
vastus medialis	W/O Exo	18.31 ± 3.00	−14.129	<0.00 **	23.02 ± 2.94	−2.158	0.038 *	31.36 ± 4.06	19.275	<0.00 **
	W/Exo	29.88 ± 8.74			24.41 ± 2.09			20.66 ± 2.45		

** p* < 0.05; ** *p* < 0.001. Data are presented as mean ± standard deviation. W/O Exo: without exoskeletons, W/Exo: with exoskeletons.

**Table 4 healthcare-13-00700-t004:** Stance phase and swing phase values during gait with and without exoskeleton.

Gait Parameter (%)		10 min	t	*p*	20 min	t	*p*	30 min	t	*p*
Stance Phase	Right W/O Exo	72.69 ± 0.48	2.737	0.01 *	71.47 ± 0.48	−2.66	0.012 *	71.11 ± 0.44	−12.109	<0.00 **
	Right W/Exo	72.09 ± 1.35			72.28 ± 1.81			74.50 ± 1.60		
	Left W/O Exo	71.62 ± 0.49	7.531	<0.00 **	70.78 ± 0.55	−13.885	<0.00 **	70.73 ± 0.52	−14.680	<0.00 **
	Left W/Exo	70.65 ± 0.12			74.18 ± 1.49			74.86 ± 1.71		
Swing Phase	Right W/O Exo	28.58 ± 0.48	3.777	0.001 *	28.04 ± 0.51	1.376	0.178	29.55 ± 0.34	10.799	<0.00 **
	Right W/Exo	27.46 ± 1.78			27.55 ± 1.98			27.26 ± 1.24		
	Left W/O Exo	29.40 ± 0.44	4.356	<0.00 **	30.43 ± 0.52	3.795	0.001 *	30.13 ± 1.77	5.079	<0.00 **
	Left W/Exo	28.29 ± 1.48			28.72 ± 2.50			23.94 ± 6.82		

** p* < 0.05; ** *p* < 0.01. Data are presented as mean ± standard deviation. W/O Exo: without exoskeletons, W/Exo: with exoskeletons.

**Table 5 healthcare-13-00700-t005:** Loading response and pre-swing values during gait with and without exoskeleton.

Gait Parameter (%)		10 min	t	*p*	20 min	t	*p*	30 min	t	*p*
Loading Response	Right W/O Exo	20.27 ± 2.21	−6.175	<0.00 **	21.79 ± 1.87	11.457	<0.00 **	23.02 ± 0.99	28.305	<0.00 **
	Right W/Exo	22.96 ± 1.08			18.51 ± 0.95			17.73 ± 0.52		
	Left W/O Exo	21.81 ± 1.87	−3.472	0.001 *	22.85 ± 1.80	10.727	<0.00 **	23.30 ± 1.15	17.772	<0.00 **
	Left W/Exo	23.29 ± 1.16			19.91 ± 0.92			18.57 ± 1.17		
Pre-Swing	Right W/O Exo	21.76 ± 1.94	2.042	0.049 *	21.27 ± 2.58	1.376	<0.00 **	22.55 ± 1.41	10.799	<0.00 **
	Right W/Exo	20.54 ± 2.85			21.82 ± 1.86			17.31 ± 0.74		
	Left W/O Exo	20.50 ± 1.58	4.356	0.148	18.56 ± 0.91	3.795	0.041 *	20.91 ± 1.14	5.079	<0.00 **
	Left W/Exo	21.27 ± 2.85			20.02 ± 0.85			18.29 ± 0.76		

** p* < 0.05; ** *p* < 0.001 Data are presented as mean ± standard deviation. W/O Exo: without exoskeletons, W/Exo: with exoskeletons.

**Table 6 healthcare-13-00700-t006:** Step length and stride length values during gait with and without exoskeleton.

Gait Parameter (cm)		10 min	t	*p*	20 min	t	*p*	30 min	t	*p*
Step Length	Right W/O Exo	72.96 ± 1.41	4.181	<0.00 **	72.11 ± 1.24	−0.279	0.782	71.53 ± 1.72	−7.191	<0.00 **
	Right W/Exo	71.57 ± 1.80			72.22 ± 1.49			75.11 ± 2.65		
	Left W/O Exo	75.06 ± 1.24	12.568	<0.00 **	71.63 ± 1.45	−0.148	0.883	72.08 ± 1.27	−11.046	<0.00 **
	Left W/Exo	72.12 ± 1.32			71.69 ± 1.55			75.92 ± 1.55		
Stride Length	W/O Exo	148.03 ± 2.62	8.366	<0.00 **	143.74 ± 1.88	−0.287	0.776	143.61 ± 2.67	−9.573	<0.00 **
	W/Exo	143.70 ± 2.82			143.91 ± 2.35			151.04 ± 3.92		

** *p* < 0.001. Data are presented as mean ± standard deviation. W/O Exo: without exoskeletons, W/Exo: with exoskeletons.

## Data Availability

Data is available from the corresponding author upon reasonable request.

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
