# Peer review of "Effect of Wearable Exoskeleton Robots on Muscle Activation and Gait Parameters on a Treadmill: A Randomized Controlled Trial"

_healthcare, 2025, doi:10.3390/healthcare13070700_

Round 1
Reviewer 1 Report
Comments and Suggestions for Authors
Effect of Wearable Exoskeleton Robots on Muscle Activation 2 and Gait Parameters on a Treadmill:
Comment 1: The title can be modified. It’s a bit difficult to understand whether it is a review or research article
Comment 2: In abstract; in method section ‘Participants engaged in a randomized 16 sequence gait experiment on’. Number of participants can be mentioned.
Comment 3: Can you check line 36
Comment 4: ‘The electrodes were positioned according to 125 to standard EMG placement protocols’ Can some references be given for this statement?
Comment 5: ‘LEXO-V Wearable Robot’ Some registered or trademark can be added.
Comment 6: initially a discussion on this robot controlled device can be discussed. Int. J. Precis. Eng. Manuf. 17, 957–964; https://doi.org/10.1016/B978-0-12-819728-8.00092-9; Sci Rep 13, 7269 (2023).
Comment 7: Is there any limitation related with the study? If it is so, can be discussed.
Comment 8: Highlight can be added
Comment 9: The exoskeleton robot on muscle activation/mobility checking has been reviewed before. The unique/novelty of this work should be explicitly included in the introduction section.
Comment 9: If all authors are from same university; why different superscripts such as 1,2, 3, 4 have been used in the author page
Comments on the Quality of English LanguageIt's fine
Author Response
Comment 1: The title can be modified. It’s a bit difficult to understand whether it is a review or research article
Response 1: Thank you for your valuable feedback. The title has been modified for clarity. Initially, it was 'Effect of Wearable Exoskeleton Robots on Muscle Activation and Gait Parameters on a Treadmill.' To better indicate that this is a research article rather than a review, it has been changed to Effect of Wearable Exoskeleton Robots on Muscle Activation and Gait Parameters on a Treadmill: A Randomized Controlled Trial
Comment 2: In abstract; in method section ‘Participants engaged in a randomized 16 sequence gait experiment on’. Number of participants can be mentioned.
Response: 2 Thank you for the suggestion. I’ve made the adjustment as follows: 'Thirty-six participants engaged in a randomized sequence gait experiment on a treadmill for 30 minutes
Comment 3: Can you check line 36
Response 3: Thank you for your observation regarding line 36. It appears We made an incorrect notation in that section, which we have since corrected and removed. We appreciate your attention to detail
Comment 4: ‘The electrodes were positioned according to 125 to standard EMG placement protocols’ Can some references be given for this statement?
Response 4: Thank you for your valuable feedback. We have now explicitly included the unique and novel aspects of our work in the introduction section as per your suggestion. We appreciate your time and effort in reviewing our manuscript
Comment 5: ‘LEXO-V Wearable Robot’ Some registered or trademark can be added.
Response 5: Thank you for your insightful comment. We have now included the registered trademark for the 'LEXO-V Wearable Robot' as suggested. We appreciate your attention to detail and your helpful feedback
Comment 6: initially a discussion on this robot controlled device can be discussed. Int. J. Precis. Eng. Manuf. 17, 957–964; https://doi.org/10.1016/B978-0-12-819728-8.00092-9; Sci Rep 13, 7269 (2023).
Response 6: Thank you for your suggestion. I will begin the discussion by addressing the robot-controlled device, the exoskeleton robot, and its potential benefits
Comment 7: Is there any limitation related with the study? If it is so, can be discussed.
Response 7: Thank you for your helpful feedback. We have now included a discussion of the limitations of our study, as per your suggestion. We appreciate your valuable input in improving the clarity and depth of our work
Comment 8: Highlight can be added
Response 8: Thank you for your valuable feedback. We have now explicitly included the unique and novel aspects of our work in the introduction section as per your suggestion. We appreciate your time and effort in reviewing our manuscript
Response 8:
Comment 9: The exoskeleton robot on muscle activation/mobility checking has been reviewed before. The unique/novelty of this work should be explicitly included in the introduction section.
Response 9: Thank you for your valuable feedback. We have now explicitly included the unique and novel aspects of our work in the introduction section as per your suggestion. We appreciate your time and effort in reviewing our manuscript
Comment 10: If all authors are from same university; why different superscripts such as 1,2, 3, 4 have been used in the author page
Response 10: Thank you for your valuable feedback. We have removed the superscripts (1, 2, 3, 4) and standardized the author page as per your suggestion. We appreciate your attention to detail and helpful comments

Reviewer 2 Report
Comments and Suggestions for Authors
The study addresses an important topic in rehabilitation and mobility assistance, providing valuable insights into the effects of exoskeletons on muscle activation and gait parameters. However, the paper's quality can be improved after considering the following comments:
- Introduction section: The paper discusses how exoskeletons help with walking and muscle activity but does not clearly explain what new contribution this study provides that previous research has not already covered.
- Participants: The study focuses only on young, healthy adults, which limits generalizability. This limitation should be further discussed in the conclusion.
- Figure 2, 3: With those Figures, it is confusing how the EMG attaches to the Subject with the type of clothes. It leads to an effect on measurement results. Please clarify.
- Males and females may have different muscle activation patterns, fatigue resistance, and gait biomechanics due to differences in muscle mass, hormonal influences, and neuromuscular control. The study does not analyze whether the exoskeleton affects males and females differently. Including this analysis could provide more valuable insights.
- About the exoskeleton: The paper could mention whether the exoskeleton was adjusted to fit individual participants or if any gender-based adjustments were needed. Please clarify the content on lines 153-154.
- Experimental procedure: Since muscle adaptation occurs over time. Why did the authors choose the walking time as 30 minutes? Is it long enough? Please provide further discussion.
Typos:
- Error citations at line 36.
- Acknowledgments still use the template content.
Author Response
Comments 1: Introduction section: The paper discusses how exoskeletons help with walking and muscle activity but does not clearly explain what new contribution this study provides that previous research has not already covered.
Response 1: Thank you for your valuable feedback. Based on your suggestion, we have clarified the novel contribution of our study compared to previous research.
Comment 2: Participants: The study focuses only on young, healthy adults, which limits generalizability. This limitation should be further discussed in the conclusion.
Response 2: Thank you for your insightful comment. We mentioned gender-related differences in muscle activation, fatigue resistance, and gait biomechanics at the beginning of the Discussion section
Comment 3 Figure 1, 2: With those Figures, it is confusing how the EMG attaches to the Subject with the type of clothes. It leads to an effect on measurement results. Please clarify.
Response 3 : We did not obtain consent from the participants for photographing them, so the current images were taken by the researcher. To improve clarity, we retake the photos and include them for better understanding
Comment 4: Males and females may have different muscle activation patterns, fatigue resistance, and gait biomechanics due to differences in muscle mass, hormonal influences, and neuromuscular control. The study does not analyze whether the exoskeleton affects males and females differently. Including this analysis could provide more valuable insights.
Response 4: Thank you for your insightful comment. We mentioned gender-related differences in muscle activation, fatigue resistance, and gait biomechanics at the beginning of the Discussion section
Comment 5: About the exoskeleton: The paper could mention whether the exoskeleton was adjusted to fit individual participants or if any gender-based adjustments were needed. Please clarify the content on lines 153-154.
Response 5: Thank you for your insightful suggestion. I have now added clarification
Comment 6: Experimental procedure: Since muscle adaptation occurs over time. Why did the authors choose the walking time as 30 minutes? Is it long enough? Please provide further discussion.
Response 6: Thank you for your thoughtful comment. I have added further discussion in the manuscript regarding the choice of a 30-minute walking duration, considering that muscle adaptation occurs over time

Reviewer 3 Report
Comments and Suggestions for Authors
1. Line 36: "[1] or [2,3], or [4–6]. See the end of the document for further details on references. Remove!
2. Lines 50-51: "Exoskeleton robots provide repetitive, precise movements that can enhance [5–7]." Sentence is not finished?
3. Line 101: "Unlike previous research..." What research? Provide appropriate references.
4. At the end of the Introduction, in the form of one paragraph, give the structure of the paper.
5. Form a subsection: "1.1 State of the art". Remove lines 59-98 from the Introduction and transfer them to State of the art. Lines 59-61: "Previous studies..." and "These studies..." Which studies? Provide appropriate references. Complete the State of the art with relevant research, in accordance with the topic of the paper.
6. Provide a reference for the device type: (i) line 143: “LEXO-V Wearable Robot”, (ii) line 160: “FREEEMG 1000”, (iii) line 166: “Optogait system”.
7. Figs. 1-3 and Tables 1-6: are not mentioned anywhere in the text, so link them. (The question arises why Figs. and Tables that are not mentioned in the text are given?)
8. Figs 2a and 2b: reduce the size vertically.
9. Fig. 4: explain and link to the corresponding part of the paper.
10. The results within the section of the same name are described briefly, while no table is mentioned in the results. The problem is that the results are described in detail in the Discussion, where they have no place in their current form, especially considering that the discussion extends over more than 4 pages! I find the Discussion too long and very difficult to read. Therefore, I suggest that the part of the discussion related to the results be moved within the Results section, in accordance with the appropriate tables individually (under each one).
11. Discussion: I do not see that all the references given in the discussion are related to the text – no comparison? Since these are findings from this paper, the question arises as to why the authors referenced their findings. The discussion should include, among other things, a comparison of the findings with similar research previously described in the State of the art. Information on how the subjects perceive the use of exoskeleton is also important. Finally, devote a paragraph to the advantages and disadvantages of using exoskeleton – application, autonomy, problems, limitations...
12. Line 446: Instead of “In conclusion, this study...”, end the sentence with “This study...”.
13. There are several typos and spelling errors, so I ask the authors to thoroughly review the manuscript before submitting a new version of the paper.
See the comments to the authors.
Author Response
Comment 1. Line 36: "[1] or [2,3], or [4–6]. See the end of the document for further details on references. Remove!
Response 1: Thank you for your observation regarding line 36. It appears We made an incorrect notation in that section, which we have since corrected and removed. We appreciate your attention to detail
Comment 2. Lines 50-51: "Exoskeleton robots provide repetitive, precise movements that can enhance [5–7]." Sentence is not finished?
Response 2: Thank you for pointing that out. I have now completed the sentence for clarity.
Comment 3. Line 101: "Unlike previous research..." What research? Provide appropriate references.
Response 3: Thank you for your helpful comment. I have now added appropriate references to support the statement "Unlike previous research..." on line 101.
Comment 4: At the end of the Introduction, in the form of one paragraph, give the structure of the paper.
Response 4: Thank you for your valuable suggestion. We have now added a paragraph at the end of the Introduction to outline the structure of the paper.
Comment 5. Form a subsection: "1.1 State of the art". Remove lines 59-98 from the Introduction and transfer them to State of the art. Lines 59-61: "Previous studies..." and "These studies..." Which studies? Provide appropriate references. Complete the State of the art with relevant research, in accordance with the topic of the paper.
Response 5: Thank you for your suggestion. I have now formed a separate subsection titled "1.1 State of the Art" and moved lines 59-98 from the Introduction to this new section. Additionally, I have included appropriate references for "Previous studies..." and "These studies..." to provide the necessary context. State-of-the-Art section has been completed with relevant research aligned with the topic of the paper.
Comment 6. Provide a reference for the device type: (i) line 143: “LEXO-V Wearable Robot”, (ii) line 160: “FREEEMG 1000”, (iii) line 166: “Optogait system”.
Response 6: Thank you for your feedback. I have now added the appropriate references for the device types
Comment 7: Figs. 1-3 and Tables 1-6: are not mentioned anywhere in the text, so link them. (The question arises why Figs. and Tables that are not mentioned in the text are given?)
Response 7: Thank you for your valuable feedback. I have now linked all the figures and tables (Figs. 1-3 and Tables 1-6) within the text.
Comment 8. Figs 2a and 2b: reduce the size vertically.
Response 8: Thank you for your valuable feedback. I adjust the figures by reducing their vertical size as requested.
Comment 9: Fig. 4: explain and link to the corresponding part of the paper.
Response 9: Thank you for your suggestion. I have now explained Fig. 4 and linked it to the corresponding part of the paper to ensure consistency.
Comment 10: The results within the section of the same name are described briefly, while no table is mentioned in the results. The problem is that the results are described in detail in the Discussion, where they have no place in their current form, especially considering that the discussion extends over more than 4 pages! I find the Discussion too long and very difficult to read. Therefore, I suggest that the part of the discussion related to the results be moved within the Results section, in accordance with the appropriate tables individually (under each one).
Response 10: Thank you for your helpful feedback. I have moved part of the discussion related to the results into the Results section, aligning it with the appropriate tables.
Comment 11: Discussion: I do not see that all the references given in the discussion are related to the text – no comparison? Since these are findings from this paper, the question arises as to why the authors referenced their findings. The discussion should include, among other things, a comparison of the findings with similar research previously described in the State of the art. Information on how the subjects perceive the use of exoskeleton is also important. Finally, devote a paragraph to the advantages and disadvantages of using exoskeleton – application, autonomy, problems, limitations...
Response 11: Thank you for your valuable feedback. I have revised the Discussion to include a comparison of the findings with similar research
Comment 12. Line 446: Instead of “In conclusion, this study...”, end the sentence with “This study...”.
Response 12: Thank you for your thoughtful suggestion. I have revised the sentence as requested.
Comment 13: There are several typos and spelling errors, so I ask the authors to thoroughly review the manuscript before submitting a new version of the paper.
Response 13: Thank you for pointing that out. I will thoroughly review the manuscript to correct any typos and spelling errors before submitting the revised version of the paper

Reviewer 4 Report
Comments and Suggestions for Authors
The aim of this study was to evaluate the influence of a wearable exoskeleton robot on gait on a treadmill. The study is of interest, although limited to healthy adult subjects, as it provides insight into certain aspects of walking by comparing the values of gait parameters obtained when walking without an exoskeleton with those obtained when walking with an exoskeleton. The study as a whole is quite clear, but the experimental procedure needs some detailed explanation in order to have a full understanding of the results and discussion. Below are my suggestions:
- Page 3 - Line 117 – 118. The reference to the University is unclear: it is not clear what “D graduate University” is. Please specify this reference better and, if it exists, also provide a reference link.
- Although it is understandable to link the tables to their description, it would be better to state explicitly which table is referred to in the text and especially in the description of the results.
- Page 5 – Line 168 – 169. It is unclear where the participants walked 7 metres: on the treadmill during the 30 minutes? Optogait measurements were taken at 10, 20 and 30 minutes (as shown in Tables 4, 5, 6). Please describe the walking protocol in detail.
- Page 5 – Section “2.4. Electromyography” and section “2.5. Optogait”. It would be useful to indicate which parameters are measured by the EMG and the Optogait system, including their units of measurement, which should then be specified in the results tables.
- Page 7 – Line 210 – Table 4. It is unclear why the values for the “Right W/O Exo” and “Left W/ Exo” are repeated for each walking phase (Stance Phase and Swing Phase): perhaps it should be the “Right W/O Exo” and “Right W/ Exo” and similarly for the left? Please specify and clarify the meaning of the parameters in the table.
- Page 8 – Line 221 – Table 5. The same considerations that were expressed for Table 4.
- Page 8 – Line 232 – Table 6. The same considerations that were expressed for Table 4.
- Page 10 – Line 312 - 313. Are we talking about MVC or gastrocnemius fatigue? I don't understand the meaning of the phrase “indicating no significant change in muscle activation” at line 313: which table are you referring to? Please specify.
Author Response
Comment 1: Page 3 - Line 117 – 118. The reference to the University is unclear: it is not clear what “D graduate University” is. Please specify this reference better and, if it exists, also provide a reference link.
Response 1: I edited by deleting the word “D” and adding “reference.” We appreciate your advice. The revised version now includes DONGBANG CULTURE UNIVERSITY.
Comment 2: Although it is understandable to link the tables to their description, it would be better to state explicitly which table is referred to in the text and especially in the description of the results.
Response 2: Thank you for your suggestion. I have now explicitly indicated which table is being referred to in the text, particularly in the description of the results
Comment 3:Page 5 – Line 168 – 169. It is unclear where the participants walked 7 metres: on the treadmill during the 30 minutes? Optogait measurements were taken at 10, 20 and 30 minutes (as shown in Tables 4, 5, 6). Please describe the walking protocol in detail.
Response 3: Thank you for your feedback. I have now provided a more detailed description. I have now linked all the figures and tables (Figs. 1-3 and Tables 1-6) within the text.
Comment 4: Page 5 – Section “2.4. Electromyography” and section “2.5. Optogait”. It would be useful to indicate which parameters are measured by the EMG and the Optogait system, including their units of measurement, which should then be specified in the results tables.
Response 4: Thank you for your suggestion. I have now included the parameters measured by the EMG and the Optogait system, along with their units of measurement, and have specified these details in the results tables.
Comment 5: Page 7 – Line 210 – Table 4. It is unclear why the values for the “Right W/O Exo” and “Left W/ Exo” are repeated for each walking phase (Stance Phase and Swing Phase): perhaps it should be the “Right W/O Exo” and “Right W/ Exo” and similarly for the left? Please specify and clarify the meaning of the parameters in the table.
Response 5: Thank you for your comment. I correct the description in Table 4
Comment 6: Page 8 – Line 221 – Table 5. The same considerations that were expressed for Table 4.
Response 6: Thank you for your comment. I correct the description in Table 5
Comment 7: Page 8 – Line 232 – Table 6. The same considerations that were expressed for Table 4.
Response 7: Thank you for your comment. I correct the description in Table 4
Comment 8: Page 10 – Line 312 - 313. Are we talking about MVC or gastrocnemius fatigue? I don't understand the meaning of the phrase “indicating no significant change in muscle activation” at line 313: which table are you referring to? Please specify.
Response 8: Thank you for your comments. I have revised the text to clarify that we are referring to gastrocnemius fatigue, and I have moved the results to Table 2.

Round 2
Reviewer 1 Report
Comments and Suggestions for Authors
Comment 1: They have not fully worked on the suggested comments such as comment 6
Comment 2: Check ref. 39 and 40
Comment 3: year for ref. 31 is wrong.
Comment 3: Check your manuscript thoroughly to remove such mistakes
Comments on the Quality of English LanguageNot needed
Author Response
Comment 1: They have not fully worked on the suggested comments such as comment 6
Response 1: We have addressed the reviewer’s comment by adding an initial discussion on the LEXO-V Wearable Robot in Section 5.1. This section now provides a detailed explanation of the device’s design and function.
First Paragraph: We introduced the LEXO-V Wearable Robot as a non-powered strength-assist exoskeleton designed to support prolonged standing and walking. Unlike motorized exoskeletons, it utilizes a passive assist mechanism to enhance stability and reduce muscle strain during movement. Additionally, we highlighted its role in minimizing fatigue accumulation and excessive muscle activation by providing external support to the waist and thighs.
Second Paragraph: We expanded on the structural and functional aspects of the exoskeleton, detailing its lightweight and robust backplate, Kevlar-based wire system with a tensile strength exceeding 150 kg, and adaptive pulley stop mechanism. These design elements ensure stability, prevent restriction of movement, and provide dynamic support based on user motion. We also emphasized its ergonomic advantages in industrial and occupational settings, enhancing work efficiency and reducing biomechanical stress.
Comment 2: Check ref. 39 and 40
Response 2: We have checked references 39 and 40 and corrected any inaccuracies.
Comment 3: year for ref. 31 is wrong.
Response 3: The year for reference 31 has been corrected.
Comment 4: Check your manuscript thoroughly to remove such mistakes
Response 4: We have thoroughly reviewed the manuscript to identify and eliminate such errors.
Reviewer 3 Report
Comments and Suggestions for Authors
1. After revision, from the Introduction it is no longer clear what the goal of this research is.
2. The previous comment is not implemented: “At the end of the Introduction, in the form of one paragraph, give the structure of the paper.” Solution – for example, “The first section presents the motivation and goal of the research; based on the available literature, the second section presents the state-of-the art; the third section is focusing…”
3. State of the art: before subsection 2.1, in the form of one sentence, give what is analyzed within the state of the art – for example: “Based on the available literature, the literature review includes three groups of problems: (i) maximum voluntary contraction with wearable exoskeleton, (ii) muscle fatigue analysis with wearable exoskeleton and (iii) changes in gait parameters associated with exoskeleton”. Supplement each of these subsections with a few more relevant papers, if any. After that, create a subsection “2.4 Summary”, and briefly conclude this overview of the state (what has been done in the field), as well as what is being considered further, which represents the topic of this paper – the goal and contribution.
4. After the first mention of a Figure or Table in the paper, immediately after the end of the paragraph, a Figure or Table should be given, which is not the case here, so take care of that.
5. Section “6. Conclusion” and Sections “7. Limitations and Future Research”: Within Section 6, as well as Section 7, future research is mentioned, so the question arises why the text from Section 7 was not inserted and modified within Section 6? I do not see the need for Section 7 as a separate section. I suggest modifying Sections 6 and 7 as one section – Conclusion.
Author Response
Comment 1: After revision, from the Introduction it is no longer clear what the goal of this research is.
Response 1:
To clarify the research goal, we have revised the Introduction as follows:
First Paragraph:
- This paragraph emphasizes the importance of investigating long-term exoskeleton use in real-world work environments.
- We specify the need to examine muscle fatigue accumulation, variations in muscle activation, and gait pattern adaptations to understand how exoskeletons affect the body over time.
- This provides clear motivation for the study, linking it to both scientific inquiry and practical applications in occupational and rehabilitation settings.
Second Paragraph:
- We explicitly state how this research contributes to optimizing exoskeleton design and application strategies, particularly in industrial and rehabilitation contexts.
- By understanding the long-term effects of exoskeleton use on muscles and gait, we aim to propose technical solutions that minimize negative impacts and enhance user comfort and safety.
- The paragraph concludes by reinforcing the practical significance of the study, such as reducing physical strain, lowering injury risks, and improving rehabilitation outcomes.
Through these revisions, we have made the research goal more explicit and highlighted its relevance to both scientific advancement and practical implementation.
Comment 2: The previous comment is not implemented: “At the end of the Introduction, in the form of one paragraph, give the structure of the paper.” Solution – for example, “The first section presents the motivation and goal of the research; based on the available literature, the second section presents the state-of-the art; the third section is focusing…”
Response 2: We added the third section, titled "Evaluating Exoskeletons for Industrial Use: Purpose and Impact", which directly addresses the purpose and impact of the research, contributing to the context and significance of the study within industrial and rehabilitation settings.
Comment 3: State of the art: before subsection 2.1, in the form of one sentence, give what is analyzed within the state of the art – for example: “Based on the available literature, the literature review includes three groups of problems: (i) maximum voluntary contraction with wearable exoskeleton, (ii) muscle fatigue analysis with wearable exoskeleton and (iii) changes in gait parameters associated with exoskeleton”. Supplement each of these subsections with a few more relevant papers, if any. After that, create a subsection “2.4 Summary”, and briefly conclude this overview of the state (what has been done in the field), as well as what is being considered further, which represents the topic of this paper – the goal and contribution.
Response 3: Thank you for your comments. Based on your feedback, I have made the following revisions to the "State of the Art" section to address your concerns:
2.1 Maximum Voluntary Contraction with Wearable Exoskeleton:
In this subsection, I have expanded the discussion by including relevant studies on the reduction of muscle activation and redistribution of loads across muscle groups, specifically related to MVC. I also clarified the importance of examining muscle fatigue and adaptation over longer walking sessions (10, 20, and 30 minutes), which were not commonly addressed in previous research.
2.2 Muscle Fatigue Analysis with Wearable Exoskeleton:
In this subsection, I further elaborated on the role of wearable exoskeletons in reducing muscle fatigue, adding more recent studies and examples of fatigue reduction in lower back and hip muscles. The discussion now emphasizes how exoskeletons contribute to fatigue delay and energy optimization during extended walking sessions, addressing gaps in existing research.
2.3 Changes in Gait Parameters Associated with Exoskeleton:
Here, I expanded on gait parameter changes by including additional studies on exoskeleton effects on step length, stride length, and phase duration. I highlighted the gap in existing studies by emphasizing the combined analysis of muscle activation, fatigue, and gait parameters within a single experimental framework.
2.4 Research Summary and Implications:
This section now provides a concise summary that synthesizes the findings from the previous subsections, emphasizing the gaps in existing research (e.g., the lack of comprehensive studies analyzing muscle activation, fatigue, and gait parameters simultaneously). It also presents the contribution of this study, which integrates all three factors into one extended walking session, providing a more holistic understanding of the effects of wearable exoskeletons.
Comment 4: After the first mention of a Figure or Table in the paper, immediately after the end of the paragraph, a Figure or Table should be given, which is not the case here, so take care of that.
Response 4: I understand your comment, and while this may not apply to my paper, I will take note of it. Thank you!
Comment 5: Section “6. Conclusion” and Sections “7. Limitations and Future Research”: Within Section 6, as well as Section 7, future research is mentioned, so the question arises why the text from Section 7 was not inserted and modified within Section 6? I do not see the need for Section 7 as a separate section. I suggest modifying Sections 6 and 7 as one section – Conclusion.
Response 5:
Thank you for your insightful comments. Based on your suggestion, I have merged Section 6 (Conclusion) and Section 7 (Limitations and Future Research) into a single Conclusion section.

Reviewer 4 Report
Comments and Suggestions for Authors
Dear authors, I think the manuscript still needs further explanation in some places, so here are my comments:
- Page 6 and following. – Section 4. Results. With regard to my previous comment number 2, it would be appropriate to indicate which table is also referred to in the first paragraphs describing the results: line 209 for Table 2, line 242 for Table 3, line 265 for Table 4, line 284 for Table 5, line 301 for Table 6.
- Page 6 – Section 3.5. Optogait. Regarding my previous comment number 3, I had said that it was not clear where the participants were walking 7 metres for the Optogait system. Now, unfortunately, I still don't understand and the protocol seems even more confusing. For the EMG measurement it says that the subjects are walking at 3 m/s (page 4, line 134), then for the Optogait recording it says that the speed is 1.7 km/h (page 6, line 193) (i.e. 0.47 m/s). So there are two walking sessions, one for the EMG and one for the Optogait? It also says that the participants were instructed to walk along a 7-meter-long bar: but before the treadmill walk? Please specify the protocol of the walk and the number of sessions.
- Page 6 - Section 3.5. Optogait. With regard to my previous comment number 4, concerning the Optogait system, the authors have not responded: they have added the character “%” in Tables 4 and 5 and “cm” in Table 6, but have not given a comprehensible explanation of what the magnitude of the reported measurements is. A brief description of these parameters would be helpful. Furthermore, for the parameters reported in %, it would be helpful to specify what the % refers to (e.g. in relation to the duration of the total gait cycle or to a phase of it), and for the parameters reported in cm, it would be helpful to report the usual values for a normal gait.
- Page 7 - Line 226-229. I think the description of the results for GM is wrong. I can see from Table 2 that the MVC of the gastrocnemius muscle is lower at 10- and 20-minutes during NON-exoskeleton condition, while it is higher at 30 minutes. Please check.
- Page 9 – Line 297. The sentence “In the loading response phase, the values were lower at the 10-minute mark and” is not clear. What values are being referred to?
Author Response
Comment 1: Page 6 and following. – Section 4. Results. With regard to my previous comment number 2, it would be appropriate to indicate which table is also referred to in the first paragraphs describing the results: line 209 for Table 2, line 242 for Table 3, line 265 for Table 4, line 284 for Table 5, line 301 for Table 6.
Response 1: Thank you for your comment. I have updated the manuscript to include the appropriate references to the tables in the first paragraphs describing the results.
Comment 2: Page 6 – Section 3.5. Optogait. Regarding my previous comment number 3, I had said that it was not clear where the participants were walking 7 metres for the Optogait system. Now, unfortunately, I still don't understand and the protocol seems even more confusing. For the EMG measurement it says that the subjects are walking at 3 m/s (page 4, line 134), then for the Optogait recording it says that the speed is 1.7 km/h (page 6, line 193) (i.e. 0.47 m/s). So there are two walking sessions, one for the EMG and one for the Optogait? It also says that the participants were instructed to walk along a 7-meter-long bar: but before the treadmill walk? Please specify the protocol of the walk and the number of sessions.
Response 2:
- Walking Protocol Clarification:
In the experimental procedure, participants walked on the treadmill at a constant speed of 1.7 km/h for 30 minutes. The study had two conditions: walking with and without exoskeleton. These conditions were alternated within the same session to ensure identical walking parameters for comparison. Both EMG and Optogait measurements were collected simultaneously throughout the entire session. - Optogait System Use:
The Optogait system, consisting of a 7-meter-long sensor bar, was positioned along the treadmill to measure gait parameters. Data were recorded over a 3-meter segment of the treadmill belt, with the first and last 2 meters excluded to avoid acceleration and deceleration effects. This ensured that the measurements only captured steady-state walking conditions. - Clarification of Measurement Timing:
The measurements for both EMG and Optogait were taken every 10 minutes during the 30-minute walking session. This allowed us to assess changes in muscle activation, fatigue, and gait parameters over time. The protocol involving both measurements was completed during a single walking session.
Comment 3: Page 6 - Section 3.5. Optogait. With regard to my previous comment number 4, concerning the Optogait system, the authors have not responded: they have added the character “%” in Tables 4 and 5 and “cm” in Table 6, but have not given a comprehensible explanation of what the magnitude of the reported measurements is. A brief description of these parameters would be helpful. Furthermore, for the parameters reported in %, it would be helpful to specify what the % refers to (e.g. in relation to the duration of the total gait cycle or to a phase of it), and for the parameters reported in cm, it would be helpful to report the usual values for a normal gait.
Response 3:
MVC (Maximum Voluntary Contraction): We have now clarified in Section 4.4 that MVC is expressed as a percentage of the maximum EMG signal recorded during a reference contraction. Each participant performed an MVC test prior to the walking trials, and the peak EMG amplitude from this test was used as a normalization factor. The EMG signals recorded during the walking trials were then expressed as a percentage of this MVC value (%MVC), allowing for standardized comparisons across participants and conditions.
Muscle Fatigue Assessment: We assessed muscle fatigue using both the root mean square (RMS) amplitude and the median frequency (MF) shift of the EMG signal over time. A progressive decrease in MF and an increase in RMS amplitude indicate muscle fatigue. RMS values are reported in microvolts (µV), and MF is expressed in Hertz (Hz). Muscle fatigue was quantified at 10, 20, and 30 minutes during the walking trials.
Optogait Parameters: Regarding the gait parameters measured using the Optogait system, we have included the following details:
- For percentages (%): The values represent the proportion of the total gait cycle occupied by each respective phase. Specifically, stance phase and swing phase are expressed as a percentage of the total gait cycle, while loading response and pre-swing are expressed as a percentage of the stance phase and, therefore, as part of the total gait cycle.
- For parameters in centimeters (cm): For step length and stride length, we have included reference values for normal gait based on existing literature. In healthy adults, typical step length ranges from 60 to 80 cm, and stride length ranges from 120 to 160 cm. These reference values allow for a comparative interpretation of the gait characteristics observed in our study.
Comment 4: Page 7 - Line 226-229. I think the description of the results for GM is wrong. I can see from Table 2 that the MVC of the gastrocnemius muscle is lower at 10- and 20-minutes during NON-exoskeleton condition, while it is higher at 30 minutes. Please check.
Response 4:
Thank you for pointing that out. I have revised the description of the results for the gastrocnemius muscle (GM) to accurately reflect the data.
Comment 5: Page 9 – Line 297. The sentence “In the loading response phase, the values were lower at the 10-minute mark and” is not clear. What values are being referred to?
Response 5:
Thank you for your feedback. I have revised the sentence to clarify the values being referred to.
